# Implication of Adult Hippocampal Neurogenesis in Alzheimer’s Disease and Potential Therapeutic Approaches

**DOI:** 10.3390/cells11020286

**Published:** 2022-01-15

**Authors:** Hesham Essa, Lee Peyton, Whidul Hasan, Brandon Emanuel León, Doo-Sup Choi

**Affiliations:** 1Department of Molecular Pharmacology and Experimental Therapeutics, Mayo Clinic College of Medicine, 200 First Street SW, Rochester, MN 55905, USA; essa.hesham@mayo.edu (H.E.); peyton.lee@mayo.edu (L.P.); hasan.whidul@mayo.edu (W.H.); leon.brandon@mayo.edu (B.E.L.); 2Department of Psychiatry and Psychology, Mayo Clinic College of Medicine and Science, Rochester, MN 55905, USA

**Keywords:** Alzheimer’s disease, neurogenesis, hippocampus, mitochondria, oxygen species, amyloid protein

## Abstract

Alzheimer’s disease is the most common neurodegenerative disease, affecting more than 6 million US citizens and representing the most prevalent cause for dementia. Neurogenesis has been repeatedly reported to be impaired in AD mouse models, but the reason for this impairment remains unclear. Several key factors play a crucial role in AD including Aβ accumulation, intracellular neurofibrillary tangles accumulation, and neuronal loss (specifically in the dentate gyrus of the hippocampus). Neurofibrillary tangles have been long associated with the neuronal loss in the dentate gyrus. Of note, Aβ accumulation plays an important role in the impairment of neurogenesis, but recent studies started to shed a light on the role of *APP* gene expression on the neurogenesis process. In this review, we will discuss the recent approaches to neurogenesis in Alzheimer disease and update the development of therapeutic methods.

## 1. Introduction

Over 6.2 million Americans suffer from Alzheimer’s disease (AD) with an estimated increase of this number upwards toward 13.8 million by 2060 [1]. AD is the most common cause of dementia and costs the health care system and society an estimated 355$ billion in the US alone. This stark cost is estimated to grow to 511$ billion in 2040 [2]. AD is a debilitating and disabling disease which presents with a litany of symptoms including progressive cognition and memory loss. It may also cause personality changes, aggression, and a decline in performing daily life activities [3]. From a historical perspective, Dr Alois Alzheimer, a german physician in 1907, was able to identify brain tissue changes that was apparent in a woman who died after unusual behavior, mentality changes, and memory loss. AD is a progressive neurodegenerative disease that is known to involve three distinct yet connected mechanisms, including aggregation of intracellular tau protein with subsequent neuron loss predominantly in the hippocampus [4] and accumulation of a misfolded Amyloid Beta Protein (Aβ) [5].

The hippocampus has been heavily implicated in AD pathogenesis [6]. The hippocampus is a special brain region that is capable of neuronal regeneration, also known as neurogenesis [7]. Neurogenesis is the ability of neuronal cells to regenerate in adult brain, a find that Altman and Das discovered in guinea pigs [8]. Since then, the subject of mammalian neurogenesis has been controversial, with discrepancies between different research studies on different species. However, it has been implied that new neurons in the hippocampus plays a role in spatial memory, episodic memory, and other cognitive function [9]. This calls into question how important and relevant neurogenesis is in AD pathogenesis. Here, we briefly go over the literature currently available about the relationship between AD and neurogenesis, with an emphasis on the relationship between mitochondrial dysfunction and the effects of oxidative stress on neurogenesis. We will also review the available literature on targeting oxidative stress as a possible treatment target for AD. Additionally, we will review the relation between adult neurogenesis and AD, paying special attention to how this neurodegenerative disease alters the neurogenic process. However, in order to fully appreciate how this disease state alters neurogenesis, we will first review adult neurogenesis in a healthy context.

## 2. Alzheimer’s Disease Pathogenesis

It is empirically vital to understand the different pathogenesis mechanisms involved in Alzheimer’s disease. Alzheimer’s disease pathogenesis is defined by two major mechanisms, aggregation of intracellular tau proteins [4] and accumulation of misfolded Amyloid beta proteins [5]. The gene responsible for encoding amyloid precursor protein is located on chromosome 21, and once produced the amyloid precursor protein undergoes cleavage by two distinct pathways, the amyloidogenic pathway and the non-amyloidogenic pathway [5]. In the amyloidogenic pathway it gets cleaved by β-secretase and yields a long soluble extracellular fragment (sAPPβ) and a membrane bound c-terminal peptide termed (β-CTF or C99) containing Aβ which is released after subsequent cleavage by γ-secretase as well as APP intracellular domain (AICD) [10]. Alternatively, for the non-amyloidogenic pathway, initial cleavage by α-secretase yields a slightly shorter soluble extracellular fragment termed (sAPPα) and a C-terminal membrane bound peptide termed (α-CTF or C83) which after subsequent cleavage by γ-secretase yields an extracellular small peptide called p3 of unknown function plus AICD [11,12]. AD is also characterized by tau protein (τ) accumulation and neurofibrillary tangles (NFTs) formation [13]. Tau is a microtubule associated protein that, when hyperphosphorylated in Alzheimer disease it forms paired protein filaments in the cell cytoplasm termed neurofibrillary tangles [14]. Another factor that is thought to play a role in AD pathogenesis is neuroinflammation [15]. Neuroinflammation is a rather equivocal term but in AD it has been described that Aβ accumulation can lead to activation of microglial cells [16]. This, in turn, leads to a cascade of events, including activation of astrocytes and the release of different inflammatory mediators and cytokines [17]. Recently, researchers have found that disruption of microglia function in AD is not only mediated by Aβ accumulation but by TREM2 mutations as well [18]. TREM2 functions as a cell surface receptor that interacts with different other cascades and it is involved in mediating microglial chemotaxis, phagocytosis, survival, and cellular proliferation [19]. Mazaheri et al. have described that TREM2 deficiency impairs vital microglial functions, which tightly correlates with AD and neurodegenerative diseases progression [20].

## 3. Mitochondrial Dysfunction and Oxidative Stress in AD

Various neuronal activities require large amounts of energy, and the nervous system is high energy demanding environment to which mitochondria provide a crucial energy source fulfilling role by providing ATP via oxidative phosphorylation for the maintenance and homeostasis of neuronal function [21]. Mitochondria are required for the biosynthesis of iron-sulphur and heme in neurons and participate in presynaptic transmitter synthesis in synapses [22]. Mitochondria supply essential buffering machinery to control calcium concentration during signal transduction, which is highly important for excitable cells such as neurons [23]. Neurons are known as long-living cells [24]. As a necessary center in controlling cell survival and death in variety of stress conditions, mitochondria prevent the survival of neuron via various types of stress mechanisms during the long neuronal lifespan [25]. To perform these different roles, mitochondria energetically network with one another and with other organelles to manage mitochondrial response to stress under physiological and pathological circumstances [26]. Any alteration in mitochondrial functions is linked with underlying nervous system anomalies involving neurodegenerative disorders. Mitochondrial dysfunctions trigger synaptic loss and alterations in tau and Aβ homeostasis [27]. Extensive research on brain of AD patients demonstrated mitochondria abnormalities [28]. Agreeing with this view, alteration in energy metabolism irreversibly lead up to clinical beginning of AD, dysfunction of mitochondria is recognized as a primary attribute of the disease, indicating the crucial role of mitochondria in the pathogenesis of AD [29].

### 3.1. Mitochondrial Dysfunction in Alzheimer’s Disease

A series of research projects has clearly demonstrated that mitochondrial dysfunction is a characteristic sign of neurodegenerative diseases, including AD [30,31]. The very recognized hallmark of AD in both familial and sporadic patterns is formation of senile plaques through the aggregation of extracellular amyloid-β peptide and intracellular deposition of neurofibrillary tangles (NFTs) by hyperphosphorylation of tau proteins [32]. Abnormalities of mitochondria functions result in aberrant processing of APP that leads to accumulation of Aβ-peptide and hyperphosphorylation of tau-protein by oxidative damage and energy deficit as demonstrated by Burnstock and Knight [33]. Mitochondria dysfunction has also been reported in early stage of AD that leads to energy deficiency and progression of this disease [34]. In addition, mitochondria homeostasis has largely affected in both sporadic and familiar types of AD in human samples as well as in the brain tissue of transgenic AD mouse model. Accumulations of Aβ-peptide also induced the mitochondria dysfunctions and contributes to the production of ROS via interfering with ETC in AD [35] (Figure 1). Aβ-peptide binds to the Aβ-peptide binding protein (ABAD) a neuronal mitochondria enzyme and induce apoptosis and ROS production in neurons. The binding of Aβ-peptide with ABAD induced mitochondrial and neuronal dysfunctions [36]. Inhibition the binding of Aβ-peptide with ABAD reduced the accumulation of Aβ-peptide and prevented mitochondria and neuronal dysfunctions in vitro and in vivo, which also improve the special learning memory [37].

Mitochondria play another important role in neurotransmitter synthesis and release since it is highly abundant in synapses and its dysfunctions also affect the neurotransmission. Neuroinflammation is another factor that participates in mitochondrial dysfunction. Microglia also known as the phagocytes of the brain are abundantly distributed within the brain [38]. Their main functions are clearing different pathogens and cellular debris as well as maintaining synapses in the brain providing better plasticity in neuronal circuits [39]. Aβ and oligomers are capable of recruiting and priming microglial cells by binding to them via multiple receptors thus potentiating the secretion of inflammatory cytokines such as interleukin 1, interleukin 6, TNF α, etc. [40]. Lately, however, preclinical research demonstrated that neuroinflammation accompany Aβ formation and might participate in the pathogenesis of AD. This link was made by focusing on mainly two genes encoding triggering receptor expressed on myeloid cells 2 (TREM2) and myeloid cell surface antigen (CD33) [41]. In the coming few sections, we discuss the effect of neuroinflammation and different pathogenic features of AD on mitochondria and energy metabolism.

### 3.2. Alteration in Metabolism of Energy Associates with Mitochondrial Dysfunction in AD

While the brain weighs about 2% of the total body weight, the brain uses about 20% of its energy [42]. Since the brain demands high energy, it is susceptible to altered energy metabolism and impairment in energy metabolism disrupts the brain function. Not surprisingly, alteration in energy metabolism is one of the initial and utmost reliable signs of AD [43]. Glucose is the principal substrate for the human adult brain under normal physiological circumstances, and its consumption is extensively utilized as one prime measure to evaluate metabolism of energy in the brain [44]. Data from various studies suggest that reduction in glucose consumption is a reliable sign in AD and mild cognitive impairment (MCI), a prodromal phase of AD, indicating an initial role in the progression of disease [45]. Reduction in glucose utilization was observed in cortex and hippocampus in AD brain as compared to subjects without dementia via fluoro-2-deoxyglucose positron-emission tomography (FDG-PET). The posterior cingulate cortex is also metabolically affected in the initial clinical phase of AD [46]. An increase in Aβ depositions were observed almost in every cortical area 15–25 years earlier the probable age of onset, subsequent decline in glucose metabolism selective cortical area around 5–10 years later indicating the glucose hypometabolism might be a secondary incident after Aβ aggregation in AD pathogenesis [47]. Gene set enrichment examination confirmed that disruption in mitochondrial import pathways and oxidative phosphorylation downregulation were trademarks of AD [48].

### 3.3. Reactive Oxygen Species and Their Role in Alzheimer’s Disease

Reactive oxygen species (ROS) are natural byproducts of redox reactions born from derivatives of molecular oxygen that exhibit high reactivity [49]. The greatest sources of endogenous ROS are transmembrane nicotinamide adenine dinucleotide phosphate (NADPH) oxidases (NOX) and the mitochondrial electron transport chain (ETC) [50]. Appropriate concentrations of ROS take part in cellular signaling and contribute to a myriad of cellular processes including cell proliferation, differentiation, migration, and angiogenesis [51,52,53]. For example, ROS modulate the activity of nuclear factor κ light chain enhancer of activated B cells (NF-κB), which plays a key role in innate and adaptive immune responses and inflammation. NF-κB is primarily activated by immune signals such as cytokines; however, ROS may also act on inducers of NF-κB [54,55,56,57]. Additionally, hydrogen peroxide (H_2_O_2_) can directly activate NF-κB through oxidation of inhibitor of NF-κB (IκB) [58].

ROS are characteristic of and implicated in the pathology of neurodegenerative disorders, including AD, where postmortem brain tissue of AD patients displays markers of oxidative stress [59]. Nuclear factor (erythroid-derived 2)-like 2 (NRF2) is a transcription factor that binds to the antioxidant response element (ARE), which is found in the promoter region of several genes involved in the response to oxidative stress [60]. Genetic depletion in APP/PS1 mice exacerbated cognitive deficits in spatial learning, spatial memory, working memory, and associative memory. Additionally, these cognitive deficients were associated with increased levels of Aβ, interferon-gamma (IFNγ), and microgliosis [61]. Interestingly, NRF2 activity has been shown to decrease with age in *Drosophila* and rodents [62,63].

ROS also contribute to mitochondrial energetics by regulating mitochondrial uncoupling. Generally, metabolic substrates in the mitochondria undergo oxidative phosphorylation (OXPHOS) and generate energy to produce adenosine triphosphate (ATP), the key energy currency of life. In mitochondrial uncoupling, the energy produced through these substrates is dissipated as heat through proton leak. This process normally occurs in brown adipose tissue [64]. Since the production of ROS, especially O_2_^−^, is tied to the activity of the ETC, uncoupling is a mechanism to limit oxidant production [50] (Figure 2). Various studies have reported that ROS affects nearly all kinds of macromolecules such as lipid, proteins, sugar, and nucleic acids in the brain of AD patients [65]. An enhanced level of protein oxidation markers such as carbonyls and 3-nitrotyrosine and increased glycation and glycol-oxidation indicating oxidative alterations to sugars were observed in brains of AD patients [66]. Product of lipid peroxidation such as aldehydes involving malondialdehyde (MDA), 4-hydroxynonal and acrolein were enhanced as well in different brain areas in AD and MCI [67]. Interestingly, plasmalogens, a subtype of phospholipid, is known to acts as scavenger for ROS [68]. Consequently, the lack of plasmalogens is associated with AD [69].

### 3.4. Alteration in Homeostasis of Mitochondrial Genomic in AD

Mitochondria have their own DNA known as mtDNA which encodes for 13 mitochondrial proteins of the electron transport chain complexes 2 rRNA and 22 tRNAs are essential for synthesis of mitochondrial proteins [70,71]. Though mtDNA is essential for efficient function of mitochondria, it is susceptible to oxidative harm due to its proximity to ROS production and comparative absence of DNA-defensive histones and effective mechanisms of DNA repair, which in turn leads to mutations [72]. Numerous literatures reported that patients having mtDNA mutations suffered from cognitive loss similar to that observed in AD patients [73], which suggest a vital role of mtDNA in suitable cognitive function. Participation of somatic mtDNA mutations such as common 5-kb deletion (*mtDNA Δ4977*) was found among positions 8470–8482 and 13,447–13,459 in most studies of AD, which affects the expression of complex I, III, and V in AD [74]. Previously quantitative PCR evaluation indicated deletion in frontal cortex in age associated manner [75], and a prominent 15-fold rise of this type of deletion in AD patients younger than 75 years of age [76]. An inclusive analysis of mtDNA rearrangement events observed significantly higher levels of F-type and R-type rearrangements, together with deletion, in the AD brain [77].

### 3.5. Interrupted Bioenergetics of Mitochondrial in AD

Various studies have found that defects of gene expression of metabolic pathways in AD related to mitochondria, signify direct indication for compromised mitochondria bioenergetic machinery in AD. A genome transcriptome study using laser-captured micro dissected neurons revealed significant reduction of nuclear genes encoding mitochondrial ETC subunits in the posterior cingulate cortex compared to those in the primary visual cortex, an area that is comparatively spared metabolically in AD [78]. A microarray study and quantitative RT-PCR analysis also showed a down regulation of TCA, oxidative phosphorylation, glycolytic and related pathways, in 15 out of 51 subjects of AD [79]. Moreover, recent microarray data analysis showed downregulation in nuclear-encoded, but not mitochondria-encoded, oxidative phosphorylation genes in the hippocampus of patients with AD, which however was strangely enhanced in the hippocampus of MCI patients [80]. Downregulation of complex I while enhanced mRNA expression of complexes III and IV were observed in initial and definite brain specimens of AD [81]. A bioinformatics examination of four transcriptome datasets of AD patient’s hippocampus recognized oxidative phosphorylation pathway is the most crucial pathway participated in AD [82]. Gene set enrichment investigation showed that oxidative phosphorylation of mitochondrial import pathways interruption were trademarks of AD [48].

On the other hand, proteomic and protein expression examination also verified under expressed proteins in pathways of oxidative phosphorylation as the greatest affected mechanism in the cortex of AD patients [83]. Quantitative proteomics methods revealed differentially altered mitochondria in AD brain are diverse from aging-related fluctuations signifying that abnormally regulated mitochondrial complexes such as various electron transport chain complexes and ATP-synthase are a crucial factor in pathogenesis of AD [84,85].

### 3.6. Mitophagy in Alzheimer’s Disease

Mitochondrial dysfunctions lead to leakage of electrons and enhance production of ROS that in turn lead to the damaging effect of membrane lipid, protein and nucleic acid [86]. Mitophagy, a mitochondrial specific form of autophagy, is a key mechanism for mitochondrial quality control and involves sequestration of mitochondria into autophagosomes allowing for lysosomal degradation of defective mitochondria [87]. Under physiological conditions, mitophagy is an essential key in the basal mitochondrial turnover and maintenance. Several studies have reported that abnormalities in autophagy elevate Aβ secretion, resulting in plaque formation in Alzheimer’s disease (AD). Accumulation of damaged mitochondria have been observed in the AD human brain and represents evidence for impaired mitophagy. Several molecular pathways are involved in mitophagy and its alterations leads to neurodegenerative diseases, PINK1-Parkin pathway, PINK1-independent pathways, NIP3-like protein X (NIX) dependent pathway, B-cell lymphoma 2 interacting protein 3 (BNIP3), B-cell lymphoma 2-like 13 (BCL2L13), FK506 binding protein 8 (FKBP8), prohibitin (PHB2), breast cancer gene 1 protein (NBR1), optineurin (OPTN), calcium binding and coiled-coil domain 2 (NDP52), Autophagy and Beclin 1 Regulator 1 (AMBRA1), Tax1 binding protein 1 (TAX1BP1), FUN14 domain-containing protein 1 (FUNDC1), PGAM family member 5 (PGAM5), Nipsnap Homolog 1 (NIPSNAP1), NIPSNAP2, among others. Mitophagy is subdivided into two pathways, the ubiquitin dependent pathway and Ubiquitin independent pathway [88]. One of the most important and most studied Ubiquitin dependent pathway is phosphatase and tensin homologue (PTEN)-induced putative kinase 1 (PINK1)–Parkin pathway. PINK 1 is a 581 amino acid protein and it contains a transmembrane domain and a serine/threonine kinase domain while PARKIN is a protein product of *PARK2* gene and is cytosolic E3 ubiquitin ligase [89]. Under normal conditions, PINK 1 binds to the outer membrane of mitochondria and then imported into the inner membrane by the help of translocase of outer membrane (TOM) and translocase of inner membrane (TIM23) [90]. When PINK 1 reaches the inner membrane of mitochondria, it is cleaved by two enzymes; the matrix processing peptidase (MPP) and the inner membrane protease PINK1/PGAM5 associated rhomboid such as protease [91]. However, when this process is compromised, PINK 1 stays on the outer membrane where it then phosphorylates itself, PARKIN and ubiquitin [92]. Phosphorylation and activation of PARKIN lead to ubiquitination of several mitochondrial proteins with subsequent recruitment of several autophagy receptors leading eventually to mitochondrial engulfing by autophagosomes [93]. Other parkin independent pathways via recruitment of autophagy adaptors autophagy adaptors, including optineurin (OPTN), nuclear dot protein 52 (NDP52) [94]. As discussed above, AD leads to ROS accumulation and mitochondrial dysfunction which in turn may lead to stress related mitophagy. As demonstrated by Bongarzone and colleague, increased advanced glycation end products (AGEs), which are end products of post translational modification of lipids, proteins and nucleic acids [95], and its receptor advanced glycation end product receptor (RAGE), are associated with ROS associated mitophagy [96]. Notably, these proteins are abundantly expressed in microglia, neurons and neural cells of AD patients and mouse AD models [97].

## 4. Neurogenesis

The long-held belief that neuron loss is permanent and there is no existent neural regeneration crumbled in 1967, when Altman and Das discovered that guinea pigs continue to generate newborn neurons in a process termed “neurogenesis”. Two distinct brain regions exhibited this process, namely the hippocampal sub-granular layer of the dentate gyrus, and the sub-ventricular zone lining the walls of the lateral ventricle [8]. These regions were later termed the neurogenic niches [7]. Since then, neurogenesis in the adult brain has been an area of constant debate in the scientific field. It is well established that neurogenesis persists through life with research proving it persists up to the tenth decade of life [98]. However, there is contention over neurogenesis into old age [99].

The neurogenesis process has four stages: (1) proliferation of neural stem cells (NCSs) and neural progenitor cells (NPCs); (2) fate specification of NSC/NPC along with movement commonly referred to as “migration”; (3) NPC differentiation into mature neurons; and (4) the integration of these mature neurons into synaptic networks [100] (Figure 3). Delving deeper into the neurogenesis process reveals that NSCs and NPCs are multipotent cells with high proliferation capacity. Within rodents, NSCs are composed of four subtypes (type 1, type iia, type iib, and type iii) and are found in dentate gyrus (DG) primarily in the sub-granular zone (SGZ) [100]. These subtypes differ in their proliferation rate, morphology, and protein expression profiles all of which have been characterized. Uniquely, type 1 NPCs are glia-like in appearance possessing radial processes. To add, the protein expression of the intermediate filament nestin and glial fibrillary acidic protein (GFAP) are common to these cell-types. It is important to mention that type 1 NPC have the lowest proliferation rate of all other NPC subtypes. Type iia are unique as they do not express GFAP but show expression for the T-domain transcription factor (Tbr2) and nestin, which are not radial and have an elevated proliferation rate. Type iib cells maintain Tbr2 expression. However, these cells now display commitment toward neuronal lineage through expression of microtubule-associated protein, doublecortin (DCX). Lastly, type iii cells show expression for DCX and polysialylated-neural cell adhesion molecule (PSA-NCAM) and fated for neuronal lineage eventually expression the mature neuron markers NeuN and calretinin. Functionally, NSC of the hippocampal SGZ gives rise to excitatory neurons of the DG. The DG receives afferent projections from the entorhinal cortex and efferent projections through the trisynaptic circuit to CA3 and CA1, thereby having essential roles in both learning and memory.

## 5. Neurogenesis in Alzheimer’s Disease

Due to the clinical symptoms that often accompany AD (dementia, memory loss, etc.), and since the hippocampus is one of the most significant neurogenic niches, many studies have examined the hippocampus to investigate how alterations in neurogenesis are implicated in AD. Indeed, decreased neurogenesis has been correlated with cognitive decline [101] and compromised long term memory [102]. Interestingly, neuron loss preferentially in the hippocampus is one of the most important pathological findings in AD [6]. Intriguingly, researchers have identified the CA1 region of the hippocampus to be one of the most affected regions displaying neuronal loss [103]. Despite the work suggesting that AD exacerbates the decline of neurogenesis more than physiological aging, there are only a few limited studies that clearly showed the neurogenesis within AD. NSC isolated from AD and healthy postmortem tissue and stained with Ki-67 (a nuclear protein that is associated with cellular proliferation) and Musashi1 (a stem cell marker) revealed that viability of NSC is decreased in AD subjects within the hippocampus compared to age-matched healthy controls. Moreover, NSC from AD patients reach senescence sooner than cells obtained from healthy controls obtain [104]. Musashi1 is an evolutionarily conserved RNA binding protein and stem cell marker that is an important regulator of neurogenesis by shifting the balance between self-renewal and differentiation [105]. In agreement, Ziabreva et al., 2006 reported similarly that AD patients show decreased progenitor cells using the Musashi1 marker. However, these investigators reported an increase in stem cell Nestin expression. Importantly, studies have already shown evidence of Musashi1 interaction with tau. In fact, it was observed that the formation pathologic protein was at least in part driven by aberrant interactions between Mushashi and tau in the nuclei [106]. Another contributing factor to the declination in neurogenesis is found in the relationship between intracellular neurofibrillary tangles and neuronal loss [107]. While undoubtedly the accumulation of intracellular neurofibrillary tangles plays a pathologic role in the neurogenesis process during AD, multiple studies have highlighted the accumulation of extracellular Aβ as the largest likely contributor to reduced neurogenesis in AD [108]. It has been suggested that several thousands of new neurons are being added daily to the dentate gyrus and the hippocampus [109]. Multiple studies have revealed a linear association between Aβ accumulation and decline of neurogenesis rate. Using transgenic rodent models to induce AD manifestations in mice aged 3 and 12 months, Haughey et al. showed a significant accumulation of extracellular Aβ and lower number of BrdU+ (5′-bromouridine positive) cells compared to control mice. The three months old mice exhibited no significant difference between neurogenesis rates and no significant Aβ accumulation [110]. In another study where *3xtg-AD* mice and C57BL/6 wild type mice were utilized, investigators found similar results, with a decline in neurogenesis along age, but most importantly a decrease in neurogenesis in AD mice compared to controls specially at 6,12 months of age [111].

Despite the strong possibility of negative correlation of Aβ accumulation and neurogenesis, most of the works in the available literature report partial evidence, showing a time point of accumulated Aβ when neurogenesis is reduced. On the other hand, *APP* gene expression in newborn neurons might have confounding effect. To further investigate this, utilizing bi-transgenic mice in which transgenic mutant *APP* expression was limited only to mature neurons and not expressed on progenitor cells, Michael and colleagues demonstrated that the neurogenesis process remained unaffected by Aβ accumulation [112]. Thus, Aβ accumulation alone is not sufficient to impair neurogenesis [112]. To further support this finding that *APP* gene expression may be a more impactful factor in limiting neurogenesis than Aβ accumulation, Pan and colleagues used *hAPP-I5* overexpress wild type *APP* gene and *hAPP-J20* mice which expresses the mutant type *APP* gene. While Aβ accumulation is notably higher in *hAPP-J20* mice compared to *hAPP-I5* mice, neurogenesis was similarly impaired in both models. Then, to investigate the effect of Aβ accumulation on neurogenesis, they deleted cystatin C in *hAPP-J20* mice which leads to significant reduction in Aβ accumulation levels, but the impaired neurogenesis process was not affected by reduced Aβ levels. Cystatin C (CysC) is a protease inhibitor originally identified in cerebrospinal fluid [113]. In AD, it has been found that CysC co-localize with Aβ [114]. In multiple studies it has been demonstrated that co-localization of cystatin C with Aβ in vessel walls lead to significantly more hemorrhagic events [115]. This raises the possibility that *APP* has a more profound effect in limiting neurogenesis compared to Aβ accumulation [100]. Additionally, mitochondrial dysfunction impairs neurogenesis not only in Alzheimer’s disease but in other neurodegenerative diseases [116]. As it stands, Khacko et al. have described that mitochondrial dysfunction results in the deficit of neurogenesis in both contexts of brain embryonic development in another study where they targeted the basic-helix-loop-helix transcription factor (bHLH) neurod1, which has been proven to be crucial for newborn granule cells to successfully mature and survive in the healthy adult brain [117]. Richtein et al. have described that overexpression of neurod1 not only increase neurogenesis compared to control mice, but also significantly increase mitochondrial mass [118], which further proves the fact that mitochondrial rescue can add to increase in neurogenesis and that mitochondria is definitely a possible target for rescuing neurogenesis. Consequently, suitable levels of ROS play a role in maintaining appropriate levels of neurogenesis. Mitochondria are implicated as regulators of neurogenesis, generate ROS, and ROS, when over accumulated, lead to mitochondrial dysfunction. Therefore, ROS and oxidative stress may be implicated as regulators of neurogenesis [119]. Studies indicated that reduced cellular redox status tend to favor cell proliferation whereas oxidized environments favor cell differentiation under physiological conditions [120]. Under pathological conditions, however, such as AD, mitochondrial dysfunction may tip the balance as ROS accumulate leading to oxidative stress and further impairments in mitochondrial function [121]. The most common contributor to oxidative stress is superoxide anion (O_2_^-^) [122] which is neutralized by super oxide dismutase (SOD) [123]. In a study investigating neurogenesis in mice exposed to radiation, those lacking extracellular SOD exhibited reduced numbers of BrdU+/NeuN+ cells in the subgranular zone of the hippocampus in knockout mice compared to wildtype mice [124]. In fact, regardless of the subcellular localization, SOD deficiency leads to reductions in neurogenesis in the SGZ of the hippocampus. Interestingly, however, in response to radiation, SOD deficient mice were able to maintain comparable levels of neurogenesis [125].

## 6. Neurogenesis as a Therapeutic Target in Alzheimer’s Disease

While all therapeutics that were targeting to mitigate the neurodegenerative process or decrease neuronal loss were not successful, with the wide acceptance of neurogenesis decline in AD, promoting neurogenesis has been emerged as a new AD therapeutic target. Research supports that stem cell transplantation can help decrease the neuroinflammation impact on neuronal loss and help replenish lost neurons [126]. However, the route of administration of stem cells, whether local or systemic, is still scarce and understandably depends on multiple factors such as whether the lesion is focal or multifocal with multifocal lesions (multiple sclerosis, vascular dementia and transverse myelitis), favoring a systemic administration route. The effect and reactivity of neurogenesis to inflammatory events have been widely available, as in rat models with transient brain ischemia where researchers found increase in the neurogenesis process in the subgranular zone in the dentate gyrus following the ischemic event [127]. Of course, this doesn’t come without a challenge, with studies furthermore reporting that stem cells in general can form teratocarcinoma when in vivo holding back the usage of stem cells in human patients [128]. Papers mention that an increase in survivability of adult born DGCs and decreasing competition induced cell death induced by mature DGCs can in fact restore memory precision in aged mice [129,130]. Another approach to promote neurogenesis and cell survival in rat models is P7C3 aminopropyl carbazole compound, which showed promise in improving cognitive functions in aged rats [131]. Another approach to upregulate neurogenesis was to decrease neuroinflammation, which in turn increased neurogenesis. This was achieved using minocycline, a known broad-spectrum antibiotic and anti-inflammatory drug [132]. They described how minocycline was able to decrease neuroinflammation in the hippocampus by deactivating microglial activation which is known to play a role in neurodegenerative diseases such as Alzheimer’s disease [133]. Similarly, Chakrabarti et al. have described the use of retinoid acid to enhance neurogenesis in AD patients [134]. Retinoid acid is an active natural and synthetic derivative of vitamin A which has known anti-oxidant effects. As described by Ding et al., it also attenuates microglia activation in *APP/PS1* mice. As well as it decreased Aβ plaque accumulation, and it significantly recovered deficits of learning and memory [135]. Exercise has also been described to increase cell proliferation and promote neurogenesis in adult mouse dentate gyrus [136]. In another follow up study, they described that exercise not only promote neurogenesis but also enhanced spatial learning and long-term potentiation of the dentate gyrus [137]. Several works in the available literature have also mentioned that exercise exerts its effect on neurogenesis with brain derived neurotrophic factor (BDNF) [138], which plays a key role in regulation of neurogenesis whether it is proliferation, differentiation, maturation, or plasticity [139]. Since the relationship between excessive oxidative stress and inhibited neurogenesis has been established, the ROS is a new emerging therapeutics target in AD. Massieu et al., reported that the neuronal damage induced by Aβ was prevented in rats treated with vitamin E [140]. Vitamin E exhibits antioxidant effects [141]. In another study, the higher dietary intake of vitamin E was associated with lower risk of AD [142]. Vitamin C which has antioxidant effects [143] was also shown to decrease behavioral abnormalities in AD mouse model [144]. Another possible antioxidant for treatment of AD is curcumin, a curry spice, which exhibits antioxidant effects [145]. Curcumin exerts its effects by decreasing the low-density lipoprotein oxidation and the free radicals that eventually lead to neurodegeneration [146]. Using *Tg2576* mice model of AD, curcumin reduced the plaque burden of insoluble and soluble Aβ which further shows its promise in the prevention of AD [147]. The look up for therapeutic intervention to enhance neurogenesis and alleviate the effect of neuroinflammation on neurogenesis is an ongoing challenge. Further pharmacologic and therapeutic approaches are warranted to improve cognitive functions in aged populations using adult hippocampal neurogenesis as a target.

## 7. Discussion

Adult hippocampal neurogenesis unquestionably exists in aged human brains, with its rates undeniably decreased in patients diagnosed with AD compared to healthy subjects. Additionally, the use of animal models particularly transgenic rodent models has correspondingly supported that early decrement of neurogenesis could alter the homeostatic balance within the brain and confer neuronal vulnerability within AD. Indeed, inflammation either from Aβ accumulation or dysregulated mitochondria is a central tenant influencing the rates of neurogenesis in AD. Furthermore, studies utilizing animal models and interdisciplinary techniques such as optogenetics, advance microscopic imaging, brain stimulation, or even human-derived organoids with be imperative to bettering our understanding of neurogenesis in both healthy and pathologic contexts. Greater examination and broader understanding of the unique interaction between altered adult hippocampal neurogenesis and neuroinflammation in AD will be imperative to development of novel approaches to ameliorate this neurodegenerative process.

## Figures and Tables

**Figure 1 cells-11-00286-f001:**
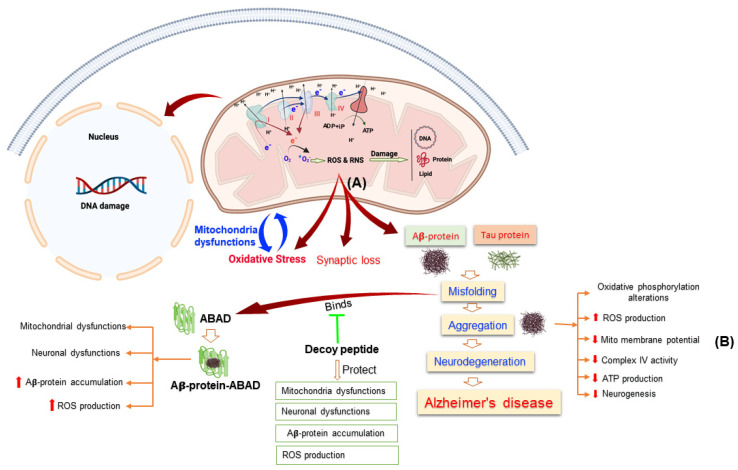
This Figure Illustrating the Alterations of Mitochondrial Functions Leads to Alzheimer’s Disease. (**A**) Mitochondria in Alzheimer’s disease induced the reactive oxygen species (ROS) and reactive nitrogen species (RNS) production that leads to oxidative stress, mitochondrial dysfunctions, damage the protein, lipid, and DNA. (**B**) Aβ-protein aggregation induced mitochondrial dysfunction through different pathways.

**Figure 2 cells-11-00286-f002:**
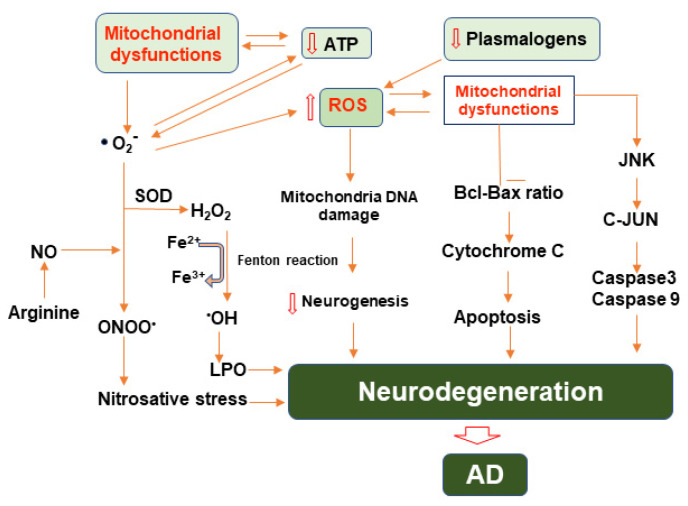
Illustration of molecular pathways involving mitochondrial dysfunction, reactive oxygen and nitrogen species, and their roles in the progression of neurodegeneration in AD. Mitochondrial dysfunction leads to enhanced generation of reactive oxygen species, primarily superoxide, which then may be converted to other ROS or combined with nitric oxide ultimately causing nitro-oxidative stress. ROS and RNS may also damage mitochondrial DNA and restrict neurogenesis. Mitochondrial dysfunction also contributes to the induction of apoptotic pathways involving Bcl/BAX and JNK ultimately exacerbating neurodegeneration.

**Figure 3 cells-11-00286-f003:**
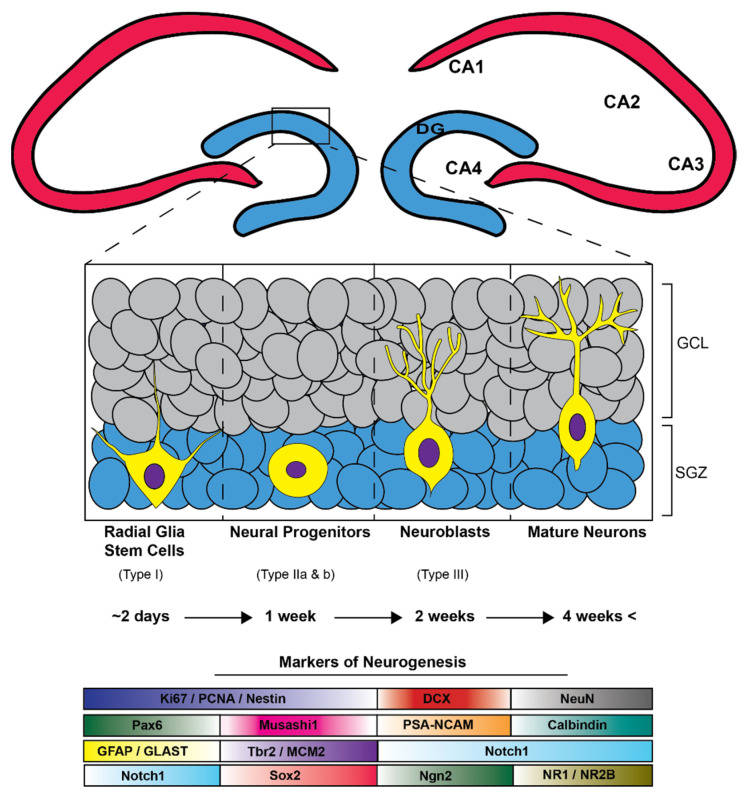
The adult hippocampal neurogenic process is hallmarked by the development and subsequent maturation of stem cells into granular cell layer neurons (GCL) in the dentate gyrus (DG). Initiation of this functional journey begins with radial-glia-like stem cells in the hippocampal subgranular zone (SGZ) which differentiate into multipotent progenitors (Type ii a and b) and then into immature neuroblasts/neurons that are subsequentially fated to mature granular cell layer neurons taking approximately four weeks beginning from radial glia-like stem cells and ending with mature GCL neurons. A subset of neurogenesis related immunohistochemical markers that can be utilized to distinguish stages and cell types of adult neurogenesis.

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
