# Peer review of "Implication of Adult Hippocampal Neurogenesis in Alzheimer’s Disease and Potential Therapeutic Approaches"

_cells, 2022, doi:10.3390/cells11020286_

Round 1

Reviewer 1 Report

The manuscript was poorly written without any cohesive theme. The authors discussed neurogenesis, oxidative stress and mitochondrial dysfunction in AD separately without any discussion how they relate to each other. What message did the authors want to deliver? There were concerns on basic scientific writing as well. Overall, the manuscript is not suitable for publication.

Most of the time, it is difficult to understand what the authors really want to say in each session. Please state the main idea in the first sentence of the paragraph and provide evidence to support the main claim with proper citations!!!

The paper needs better proof reading. Many of the sentences were difficult to understand due to grammar errors and complexity of these sentences.

It appears that the session 2.3 “Neurogenesis as a therapeutic target in AD” is purely inference without much experimental evidence in AD models.

3.1 ROS and immune response: what message did the authors want to deliver? Yes, physiological ROS could act as second messenger important in immune response. Does excessive ROS play a role in immune response? How was the second paragraph in this session supposed to be related to the immune response topic?

3.2, line 167: While uncoupling reduced ATP and ROS generation, how ROS regulates uncoupling? What is the main idea of this session?

4.1, mitochondrial dysfunction in AD: the first paragraph as a summary of mitochondrial dysfunction in AD looks very awkward without much specific information on what mitochondrial function is impaired and how. No citation was provided to support any of the claims. Among many of the Abeta-binding proteins even in the mitochondria, what is special about ABAD? The second paragraph also looks very awkward with a changing topic on some basic background on neuroinflammation. How mitochondrial dysfunction is related to neuroinflammation?

Author Response

。

Reviewer 2 Report

The authors presented an review article about " Neurogenesis, Mitochondrial Dysfunction and Oxidative Stress in Alzheimer disease". 

Altogether neurogenesis in AD is a relevant topic which may have future impact for therapeutic approaches.  The review article is well organized  and clearly presented. Minor points are: 

1) Introduction: Here the authors presented also the historical information about Alois Alzheimer which everybody in the field knows. I would guide the readers to the more important filed of neurogenesis. 

2) Secondly in AD research the main focus is on Aß or tau pathology, however all therapeutical treatments targeting both pathologies failed until now . Therefore, it is import to understand and focus on the basic mechanisms of this multi-factorial disease . In this context, the authors can then discuss those factors. 

3) Please include abbreviations in the figure legends

4) Are the energetic dysfunctions associated with an early onset or do they effect the course of the disease and how does aging promote those dysfunction increasing the risk for AD?

Author Response

We would like to thank the reviewer for the thoughtful comments and helpful suggestions. 

Reviewer #2: The authors presented a review article about " Neurogenesis, Mitochondrial Dysfunction and Oxidative Stress in Alzheimer disease". 

Altogether neurogenesis in AD is a relevant topic which may have future impact for therapeutic approaches.  The review article is well organized and clearly presented.

Minor concerns:

Q1.

Introduction: Here the authors presented also the historical information about Alois Alzheimer which everybody in the field knows. I would guide the readers to the more important filed of neurogenesis. 

Response: We appreciate the reviewer’s comment on the introduction. We revised the manuscript according to the reviewer’s suggestion, focusing more on the process of neurogenesis as follows, “The hippocampus has been heavily implicated in AD pathogenesis [1]. The hippocampus is a special brain region that is capable of neuronal regeneration also known as neurogenesis [2]. Neurogenesis is the ability of neuronal cells to regenerate in adult brain, a find that Altman & Das discovered in guineapigs [3]. Since then, the subject of mammalian neurogenesis has been controversial with discrepancies between different research and species. However, it has been implied that new neurons in the hippocampus plays a role in spatial memory, episodic memory and other cognitive function [4]. Which call in the question of how important and relevant neurogenesis in AD pathogenesis.”

Q2.

Secondly in AD research the main focus is on Aß or tau pathology, however all therapeutical treatments targeting both pathologies failed until now. Therefore, it is important to understand and focus on the basic mechanisms of this multi-factorial disease. In this context, the authors can then discuss those factors

Response: We absolutely agree with the reviewer that AD is a multi-factorial disease. In the revised manuscript, in addition to the two major factors, Ab and tau, we discussed other factors such as neuroinflammation and mitochondrial dysfunction although many are correlated.  “Another factor that is thought to play a role in Alzheimer disease is neuroinflammation [5]. Neuroinflammation is a rather equivocal term but in AD it has been described that Aβ accumulation can lead to activation of microglial cells [6]. Which in turn leads to a cascade of events, including activation of astrocytes and the release of different inflammatory mediators and cytokines [7]. Recently, Research has found that disruption of microglia function in AD is not only mediated by Aβ accumulation but with TREM2 mutations as well [8]. TREM2 functions as a cell surface receptor that interacts with different other cascades and it is involved in mediating microglial chemotaxis, phagocytosis, survival and cellular proliferation [9]. Mazaheri et al. have described that TREM2 deficiency impairs vital microglial functions which tightly correlates with AD and neurodegenerative diseases progression [10]. ” 

Q3.

Please include abbreviations in the figure legends

Response: We have included the abbreviations in our figure legends in the revised manuscript.

Q4.

Are the energetic dysfunctions associated with an early onset or do they effect the course of the disease and how does aging promote those dysfunctions increasing the risk for AD?

Response: As mentioned in our manuscript, the existence of mitochondrial and energy dysfunction has been recognized as an early molecular sign of AD that precedes behavioral and memory dysfunction. And there are literature describing that the extent of energy dysfunction is related to the extent of memory and behavioral dysfunction [11]. Trushina et al found mitochondrial dysfunction in familial Alzheimer’s disease to precede the onset of memory and neurological phenotype of the disease. Which proves that mitochondria and energy dysfunction is an underlying mechanism in AD progression [12]. Citations and description of this has been added to the mitochondria section in our revised manuscript.

Reference in the reply:

  1. Moreno-Jimenez, E.P.; Flor-Garcia, M.; Terreros-Roncal, J.; Rabano, A.; Cafini, F.; Pallas-Bazarra, N.; Avila, J.; Llorens-Martin, M. Adult hippocampal neurogenesis is abundant in neurologically healthy subjects and drops sharply in patients with Alzheimer's disease. Nat Med 2019, 25, 554-560, doi:10.1038/s41591-019-0375-9.
  2. Babcock, K.R.; Page, J.S.; Fallon, J.R.; Webb, A.E. Adult Hippocampal Neurogenesis in Aging and Alzheimer's Disease. Stem Cell Reports 2021, 16, 681-693, doi:10.1016/j.stemcr.2021.01.019.
  3. Altman, J.; Das, G.D. Postnatal neurogenesis in the guinea-pig. Nature 1967, 214, 1098-1101, doi:10.1038/2141098a0.
  4. Eriksson, P.S.; Perfilieva, E.; Bjork-Eriksson, T.; Alborn, A.M.; Nordborg, C.; Peterson, D.A.; Gage, F.H. Neurogenesis in the adult human hippocampus. Nat Med 1998, 4, 1313-1317, doi:10.1038/3305.
  5. Broussard, G.J.; Mytar, J.; Li, R.C.; Klapstein, G.J. The role of inflammatory processes in Alzheimer's disease. Inflammopharmacology 2012, 20, 109-126, doi:10.1007/s10787-012-0130-z.
  6. Akiyama, H.; Barger, S.; Barnum, S.; Bradt, B.; Bauer, J.; Cole, G.M.; Cooper, N.R.; Eikelenboom, P.; Emmerling, M.; Fiebich, B.L.; et al. Inflammation and Alzheimer's disease. Neurobiol Aging 2000, 21, 383-421, doi:10.1016/s0197-4580(00)00124-x.
  7. Hansen, D.V.; Hanson, J.E.; Sheng, M. Microglia in Alzheimer's disease. J Cell Biol 2018, 217, 459-472, doi:10.1083/jcb.201709069.
  8. Guerreiro, R.; Wojtas, A.; Bras, J.; Carrasquillo, M.; Rogaeva, E.; Majounie, E.; Cruchaga, C.; Sassi, C.; Kauwe, J.S.; Younkin, S.; et al. TREM2 variants in Alzheimer's disease. N Engl J Med 2013, 368, 117-127, doi:10.1056/NEJMoa1211851.
  9. Kleinberger, G.; Yamanishi, Y.; Suarez-Calvet, M.; Czirr, E.; Lohmann, E.; Cuyvers, E.; Struyfs, H.; Pettkus, N.; Wenninger-Weinzierl, A.; Mazaheri, F.; et al. TREM2 mutations implicated in neurodegeneration impair cell surface transport and phagocytosis. Sci Transl Med 2014, 6, 243ra286, doi:10.1126/scitranslmed.3009093.
  10. Mazaheri, F.; Snaidero, N.; Kleinberger, G.; Madore, C.; Daria, A.; Werner, G.; Krasemann, S.; Capell, A.; Trumbach, D.; Wurst, W.; et al. TREM2 deficiency impairs chemotaxis and microglial responses to neuronal injury. EMBO Rep 2017, 18, 1186-1198, doi:10.15252/embr.201743922.
  11. Dragicevic, N.; Mamcarz, M.; Zhu, Y.; Buzzeo, R.; Tan, J.; Arendash, G.W.; Bradshaw, P.C. Mitochondrial amyloid-beta levels are associated with the extent of mitochondrial dysfunction in different brain regions and the degree of cognitive impairment in Alzheimer's transgenic mice. Journal of Alzheimer's disease : JAD 2010, 20 Suppl 2, S535-550, doi:10.3233/JAD-2010-100342.
  12. Trushina, E.; Nemutlu, E.; Zhang, S.; Christensen, T.; Camp, J.; Mesa, J.; Siddiqui, A.; Tamura, Y.; Sesaki, H.; Wengenack, T.M.; et al. Defects in mitochondrial dynamics and metabolomic signatures of evolving energetic stress in mouse models of familial Alzheimer's disease. PLoS One 2012, 7, e32737, doi:10.1371/journal.pone.0032737.

Reviewer 3 Report

In this review, the authors state that they will discuss the recent approaches to neurogenesis in AD and update the development of therapeutic methods. This topic, if reviewed well, could lead to significant contribution to field, even though a comprehensive review on this topic has recently been published (https://doi.org/10.1016/j.stemcr.2021.01.019). However, a major part of this review focuses on oxidative stress and mitochondria dysfunction in AD; a topic has been extensively reviewed.

In the introduction, the authors also state that they will focus on the relationship between mitochondrial dysfunction and the effects of oxidative stress on neurogenesis; However, not even one ‘neurogenesis’ was mentioned when reviewing oxidative stress and mitochondrial dysfunction. Neither figure 2 nor figure 3 has a word of ‘neurogenesis’.

Author Response

We would like to thank the reviewer for the thoughtful comments and helpful suggestions.

Reviewer #3: In this review, the authors state that they will discuss the recent approaches to neurogenesis in AD and update the development of therapeutic methods. This topic, if reviewed well, could lead to significant contribution to field.

Minor concerns:

Q1. a major part of this review focuses on oxidative stress and mitochondria dysfunction in AD; a topic has been extensively reviewed.

Response: We agree with the reviewer’s point. Thus, in the revised manuscript, we shortened the well-known parts of oxidative stress and mitochondria dysfunction in AD. Instead, we highlight and provide new findings regarding neurogenesis, and mitochondrial dysfunction in AD.

Q2. In the introduction, the authors also state that they will focus on the relationship between mitochondrial dysfunction and the effects of oxidative stress on neurogenesis; However, not even one ‘neurogenesis’ was mentioned when reviewing oxidative stress and mitochondrial dysfunction. Neither figure 2 nor figure 3 has a word of ‘neurogenesis’.

Response: We thank you for this critical comment. We added the relationship between mitochondrial dysfunction and neurogenesis as follows “Mitochondrial dysfunction impairs neurogenesis not only in Alzheimer disease but in other neurodegenerative diseases [1]. As it shows, Khacko et al have described that mitochondrial dysfunction results in the deficit of neurogenesis in both contexts of brain embryonic development. In another study where they targeted the basic-helix-loop-helix transcription factor (bHLH) neurod1, which has been proven to be crucial for newborn granule cells to successfully mature and survive in the healthy adult brain [2]. Richtein et al have described that overexpression of neurod1 not only increase neurogenesis compared to control mice, but also significantly increase mitochondrial mass [3], which further proves the fact that mitochondrial rescue can add to increase in neurogenesis and that mitochondria is definitely a possible target for rescuing neurogenesis.”

Importantly, we added the implication of oxidative stress and mitochondrial dysfunction in neurogenesis in the Figure 2 and 3.

References in the reply:

  1. Khacho, M.; Clark, A.; Svoboda, D.S.; MacLaurin, J.G.; Lagace, D.C.; Park, D.S.; Slack, R.S. Mitochondrial dysfunction underlies cognitive defects as a result of neural stem cell depletion and impaired neurogenesis. Hum Mol Genet 2017, 26, 3327-3341, doi:10.1093/hmg/ddx217.
  2. Gao, Z.; Ure, K.; Ables, J.L.; Lagace, D.C.; Nave, K.A.; Goebbels, S.; Eisch, A.J.; Hsieh, J. Neurod1 is essential for the survival and maturation of adult-born neurons. Nat Neurosci 2009, 12, 1090-1092, doi:10.1038/nn.2385.
  3. Richetin, K.; Moulis, M.; Millet, A.; Arrazola, M.S.; Andraini, T.; Hua, J.; Davezac, N.; Roybon, L.; Belenguer, P.; Miquel, M.C.; et al. Amplifying mitochondrial function rescues adult neurogenesis in a mouse model of Alzheimer's disease. Neurobiol Dis 2017, 102, 113-124, doi:10.1016/j.nbd.2017.03.002.

Reviewer 4 Report

Essa et al summarize the recent literature dealing with the impact of neurogenesis and Alzheimer´s disease (AD). The authors focus in particular on the link between oxidative stress (and therefore on mitochondrial dysfunction leading to impaired energy homeostasis), APP processing, mitophagy, and AD. 
The review is up to date and comprehensive. It has a clear structure; the literature is reviewed in a balanced way and also basic information is mainly provided (see comment 1) helping the readership who is not so familiar with AD.
 Several reviews dealing with AD and the underlying different mechanisms exist. However, neurogenesis is often neglected. I, therefore, appreciate the efforts of the authors to summarize these topics. It is interesting for a broad readership, in particular for neuroscientists dealing with AD. In my opinion, it is in the clear scope of the journal. 
I have a few suggestions which could help to increase the impact of the paper:
1)    A graphic illustrating the APP processing would be helpful for readers non-familiar with AD summarizing the amyloidogenic and non-amyloidogenic pathway and the cleavage products as well the enzymes involved
2)     The section about mitophagy is in my opinion the weakest section. Please try to make clearer what is the mechanistic link between mitophagy and neurogenesis. Probably a table summarizing the literature and the mechanism would be helpful
3)    Figure 3 (which is in general very nice) lacks the impact of ROS on peroxisomal function. Interestingly, peroxisomes are tightly linked to Plasmalogens, which are important lipids involved in neurogenesis. Moreover, Plasmalogens (in particular the vinyl ether) are highly sensitive to ROS so a second link to plasmalogens exists. 
4)    APP processing (AICD, but also Abeta) is known to regulate several enzymes involved in lipid homeostasis linked to neurogenesis.  I would appreciate it if lipids and lipid peroxidation would not only be mentioned by the authors but would be discussed in more detail (probably in a separate chapter). Especially Plasmalogens, Sphingolipids, cholesterol but also oxidation of PUFAs and DHA and their potential impact would be helpful 
5)    It is a pity that no graphical abstract is provided. In my experience, a graphical abstract enhances the recognition of an article and the citations dramatically. If a graphical abstract is provided one could consider if the review is suitable to be highlighted.   
6)    Please enhance the section about the therapeutical approaches. I missed there some recent papers dealing with neurogenesis. Please see for example the paper of Wang and Brinton (DOI: 10.2174/156720506777632817 and some others (e.g. Sung doi:10.3390/ijms21030701)

Author Response

We would like to thank the reviewer for the thoughtful comments and helpful suggestions.

Reviewer #4: Essa et al summarize the recent literature dealing with the impact of neurogenesis and Alzheimer´s disease (AD). The authors focus on the link between oxidative stress (and therefore on mitochondrial dysfunction leading to impaired energy homeostasis), APP processing, mitophagy, and AD. The review is up to date and comprehensive. It has a clear structure; the literature is reviewed in a balanced way and basic information is mainly provided (see comment 1) helping the readership who is not so familiar with AD. Several reviews dealing with AD and the underlying different mechanisms exist. However, neurogenesis is often neglected. I, therefore, appreciate the efforts of the authors to summarize these topics. It is interesting for a broad readership, for neuroscientists dealing with AD. In my opinion, it is in the clear scope of the journal. 

Minor concerns:

Q1.   A graphic illustrating the APP processing would be helpful for readers non-familiar with AD summarizing the amyloidogenic and non-amyloidogenic pathway and the cleavage products as well the enzymes involved.

Response: Thank you for the comment and a graphical abstract has been added to our revised manuscript. 

Q2.   The section about mitophagy is in my opinion the weakest section. Please try to make clearer what is the mechanistic link between mitophagy and neurogenesis. Probably a table summarizing the literature and the mechanism would be helpful.

Response: We realize that the relationship of mitochondrial dysfunction and neurogenesis is lacking in our manuscript that is why we added this to our revised manuscript to try to develop a better relationship with neurogenesis, “Mitochondrial dysfunction impairs neurogenesis not only in Alzheimer disease but in other neurodegenerative diseases [1]. As it shows, Khacko et al have described that mitochondrial dysfunction results in the deficit of neurogenesis in both contexts of brain embryonic development. In another study where they targeted the basic-helix-loop-helix transcription factor (bHLH) neurod1, which has been proven to be crucial for newborn granule cells to successfully mature and survive in the healthy adult brain [2]. Richtein et al have described that overexpression of neurod1 not only increase neurogenesis compared to control mice, but also significantly increase mitochondrial mass [3], which further proves the fact that mitochondrial rescue can add to increase in neurogenesis and that mitochondria is definitely a possible target for rescuing neurogenesis.” 

Q3.    Figure 3 (which is in general very nice) lacks the impact of ROS on peroxisomal function. Interestingly, peroxisomes are tightly linked to Plasmalogens, which are important lipids involved in neurogenesis. Moreover, Plasmalogens (in particular the vinyl ether) are highly sensitive to ROS so a second link to plasmalogens exists. 

Response: Thank you for the suggestion. We added the impact of ROS on peroxisomal function in the Figure 3 and discussed the implication to plasmalogens in neurogenesis as follows, “Interestingly, plasmalogens, a subtype of phospholipid, is known to acts as scavenger for ROS [4]. Consequently, the lack of plasmalogens is associated with AD [5].

Q4.   APP processing (AICD, but also Abeta) is known to regulate several enzymes involved in lipid homeostasis linked to neurogenesis.  I would appreciate it if lipids and lipid peroxidation would not only be mentioned by the authors but would be discussed in more detail (probably in a separate chapter). Especially Plasmalogens, Sphingolipids, cholesterol but also oxidation of PUFAs and DHA and their potential impact would be helpful 

Response: Thank you for mentioning this approach to APP processing. While we totally agree with the reviewer on this, we think that mentioning this pathway in the separate chapter is out of scope of our review.  However, PUFA is mentioned briefly in the therapeutic approaches section in our revised manuscript.

Q5.     It is a pity that no graphical abstract is provided. In my experience, a graphical abstract enhances the recognition of an article and the citations dramatically. If a graphical abstract is provided one could consider if the review is suitable to be highlighted.

Response: We agree with the reviewer and a graphical abstract has been added to our revised manuscript.   

Q6.     Please enhance the section about the therapeutical approaches. I missed there some recent papers dealing with neurogenesis. Please see for example the paper of Wang and Brinton (DOI: 10.2174/156720506777632817 and some others (e.g. Sung doi:10.3390/ijms21030701). 

Response: Thank you for the comment and we have enhanced the therapeutic approaches section in our revised manuscript. After reading the two exemplary review papers as recommended, we cited two papers and other recent articles to discuss more on the therapeutic approaches as follow, “Another approach to upregulate neurogenesis was to decrease neuroinflammation, which in turn increased neurogenesis. This was achieved using Minocycline, a known broad-spectrum antibiotic and anti-inflammatory drug [6]. They described how Minocycline was able to decrease neuroinflammation in the hippocampus by deactivating microglial activation which is known to play a role in neurodegenerative diseases like Alzheimer disease [7]. Similarly, Chakrabarti et al have described the use of retinoid acid to enhance neurogenesis in AD patients [8]. Retinoid acid is an active natural and synthetic derivative of vitamin A. As described by Ding et al, it also attenuates microglia activation in APP/PS1 mice. As well as it decreased Aβ plaque accumulation, and it significantly recovered deficits of learning and memory [9]. Exercise has also been described to increase cell proliferation and promote neurogenesis in adult mouse dentate gyrus [10]. In another follow up study, they described that exercise not only promote neurogenesis but also enhanced spatial learning and long term potentiation of the dentate gyrus [11]. Several literatures have also mentioned that exercise exerts its effect on neurogenesis with brain derived neurotrophic factor (BDNF) [12] which plays a key role in regulation of neurogenesis whether it is proliferation, differentiation, maturation or plasticity [13]. In other reports, it has been demonstrated that multiple growth factors are implicated in the neurogenesis process like IGF-1, FGF-2 and VEGF [14]. In a study where they administered IGF-1 via intracerebroventricular pump, they found out that IGF-1 significantly increased BrdU positive nuclei compared to controls [15]. Another way to enhance neurogenesis is through dietary modification like changing the caloric intake, meal content or meal frequency [16]. One interesting finding was how omega-3 fatty acids can positively regulate neurogenesis as demonstrated in a study done by Beltz et al [17] where they found out that even few weeks manipulation of dietary intake of polysaturated fats rich in omega-3 can significantly increase BrdU positive cells in the hippocampus. The look up for therapeutic intervention to enhance neurogenesis and alleviate the effect of neuroinflammation on neurogenesis is a still ongoing challenge.”

References in the reply:

  1. Khacho, M.; Clark, A.; Svoboda, D.S.; MacLaurin, J.G.; Lagace, D.C.; Park, D.S.; Slack, R.S. Mitochondrial dysfunction underlies cognitive defects as a result of neural stem cell depletion and impaired neurogenesis. Hum Mol Genet 2017, 26, 3327-3341, doi:10.1093/hmg/ddx217.
  2. Gao, Z.; Ure, K.; Ables, J.L.; Lagace, D.C.; Nave, K.A.; Goebbels, S.; Eisch, A.J.; Hsieh, J. Neurod1 is essential for the survival and maturation of adult-born neurons. Nat Neurosci 2009, 12, 1090-1092, doi:10.1038/nn.2385.
  3. Richetin, K.; Moulis, M.; Millet, A.; Arrazola, M.S.; Andraini, T.; Hua, J.; Davezac, N.; Roybon, L.; Belenguer, P.; Miquel, M.C.; et al. Amplifying mitochondrial function rescues adult neurogenesis in a mouse model of Alzheimer's disease. Neurobiol Dis 2017, 102, 113-124, doi:10.1016/j.nbd.2017.03.002.
  4. Lessig, J.; Fuchs, B. Plasmalogens in biological systems: their role in oxidative processes in biological membranes, their contribution to pathological processes and aging and plasmalogen analysis. Curr Med Chem 2009, 16, 2021-2041, doi:10.2174/092986709788682164.
  5. Su, X.Q.; Wang, J.; Sinclair, A.J. Plasmalogens and Alzheimer's disease: a review. Lipids Health Dis 2019, 18, 100, doi:10.1186/s12944-019-1044-1.
  6. Wadhwa, M.; Prabhakar, A.; Ray, K.; Roy, K.; Kumari, P.; Jha, P.K.; Kishore, K.; Kumar, S.; Panjwani, U. Inhibiting the microglia activation improves the spatial memory and adult neurogenesis in rat hippocampus during 48 h of sleep deprivation. J Neuroinflammation 2017, 14, 222, doi:10.1186/s12974-017-0998-z.
  7. Lue, L.F.; Walker, D.G.; Rogers, J. Modeling microglial activation in Alzheimer's disease with human postmortem microglial cultures. Neurobiol Aging 2001, 22, 945-956, doi:10.1016/s0197-4580(01)00311-6.
  8. Chakrabarti, M.; McDonald, A.J.; Will Reed, J.; Moss, M.A.; Das, B.C.; Ray, S.K. Molecular Signaling Mechanisms of Natural and Synthetic Retinoids for Inhibition of Pathogenesis in Alzheimer's Disease. Journal of Alzheimer's disease : JAD 2016, 50, 335-352, doi:10.3233/JAD-150450.
  9. Ding, Y.; Qiao, A.; Wang, Z.; Goodwin, J.S.; Lee, E.S.; Block, M.L.; Allsbrook, M.; McDonald, M.P.; Fan, G.H. Retinoic acid attenuates beta-amyloid deposition and rescues memory deficits in an Alzheimer's disease transgenic mouse model. J Neurosci 2008, 28, 11622-11634, doi:10.1523/JNEUROSCI.3153-08.2008.
  10. van Praag, H.; Kempermann, G.; Gage, F.H. Running increases cell proliferation and neurogenesis in the adult mouse dentate gyrus. Nat Neurosci 1999, 2, 266-270, doi:10.1038/6368.
  11. van Praag, H.; Christie, B.R.; Sejnowski, T.J.; Gage, F.H. Running enhances neurogenesis, learning, and long-term potentiation in mice. Proceedings of the National Academy of Sciences of the United States of America 1999, 96, 13427-13431, doi:10.1073/pnas.96.23.13427.
  12. Liu, P.Z.; Nusslock, R. Exercise-Mediated Neurogenesis in the Hippocampus via BDNF. Front Neurosci 2018, 12, 52, doi:10.3389/fnins.2018.00052.
  13. Bath, K.G.; Akins, M.R.; Lee, F.S. BDNF control of adult SVZ neurogenesis. Dev Psychobiol 2012, 54, 578-589, doi:10.1002/dev.20546.
  14. Brinton, R.D.; Wang, J.M. Therapeutic potential of neurogenesis for prevention and recovery from Alzheimer's disease: allopregnanolone as a proof of concept neurogenic agent. Curr Alzheimer Res 2006, 3, 185-190, doi:10.2174/156720506777632817.
  15. Perez-Martin, M.; Cifuentes, M.; Grondona, J.M.; Lopez-Avalos, M.D.; Gomez-Pinedo, U.; Garcia-Verdugo, J.M.; Fernandez-Llebrez, P. IGF-I stimulates neurogenesis in the hypothalamus of adult rats. Eur J Neurosci 2010, 31, 1533-1548, doi:10.1111/j.1460-9568.2010.07220.x.
  16. Sung, P.S.; Lin, P.Y.; Liu, C.H.; Su, H.C.; Tsai, K.J. Neuroinflammation and Neurogenesis in Alzheimer's Disease and Potential Therapeutic Approaches. Int J Mol Sci 2020, 21, doi:10.3390/ijms21030701.
  17. Beltz, B.S.; Tlusty, M.F.; Benton, J.L.; Sandeman, D.C. Omega-3 fatty acids upregulate adult neurogenesis. Neurosci Lett 2007, 415, 154-158, doi:10.1016/j.neulet.2007.01.010.

Reviewer 5 Report

The Study by Essa et al 'Neurogenesis, Mitochondrial Dysfunction and Oxidative Stress in Alzheimer Disease' should have neurogenesis in AD as a pivotal topic. The authors write in the Abstract: "In the review, we will discuss the recent approaches to neurogenesis in Alzheimer disease and update the development of therapeutic methods.", but honestly I do not find this sentence to match the content of the manuscript.

  • If, as can be seen from the Introduction, this study intends to treat the relationship between AD and neurogenesis and how this neurodegenerative disease alters the neurogenic process, the title 'Neurogenesis, Mitochondrial Dysfunction and Oxidative Stress in Alzheimer Disease' is rather equivocal. So it definitely needs to be changed.
  • A meticulous and massive revision work is necessary and urgent. Before detailing changes to be made, I want to point out to the authors that many, many data reported lack bibliographic references. Absolutely to be included. Eg. in #paragraph 4.1 there is a lot of information reported between refs 57 and 58 that are orphans, without a paternity, i.e. without bibliographical references!!! But it is just an example!
  • The design of the study is confusing. First, I ask the authors: What is the topic they intend to address? According to what is reported in the Abstract, it seems that the spotlight is on neurogenesis in AD. Can the authors confirm this? If Yes, the title must be revised !!! After reading the Title “Neurogenesis, Mitochondrial Dysfunction and Oxidative Stress in Alzheimer Disease” I thought that these 3 different topics were covered in the Review. The Introduction reads 'Here, we briefly go over the literature currently available about the relationship between AD and neurogenesis with an emphasis on the relationship between mitochondrial dysfunction and the effects of oxidative stress on neurogenesis.' But there is no trace of what was planned in the introduction in Sections 4 and 5.
  • In paragraph 2.1, the authors first mention tau protein. They write: 'studies have already shown evidence of Musashi1 interaction with tau', but they do not introduce the tau protein to the reader, who does not necessarily know AD. The same applies to the accumulation of Ab. The authors write '… highlighted the accumulation of extracellular Aβ as the largest likely contributor to reduced neurogenesis in AD.', but again they do not take into account that a reader interested in neurogenesis, but not Alzheimer's, may not know the Abeta's role in AD. Therefore, with the risk of being repetitive, it is necessary to 'present' the disease to the 'ignorant' reader !!! I too, when I write Reviews on AD, I ask myself the problem of giving redundant information, but I realize that, even if brief, certain information on the characteristics of AD must be provided, if anything in such a way as to guide it towards the topic of interest.
  • Also in paragraph 2.2 'Is it Aβ accumulation or APP expression affecting neurogenesis?', authors mention cystatin C. Again: the authors assume that cystatin C is known to the reader as the concept of oxidative stress may be known since you study it on textbooks! If we go into the experimental detail, it is necessary to accompany the reader - whatever he/she is - to understand the study.
  • I would certainly deepen paragraph 2.3 'Neurogenesis as a therapeutic target in Alzheimer's disease'. The steps of the research in this sense and if this type of approach has passed from the laboratory to the clinic it would be very important to investigate.
  • As regards Section 3. '3. Reactive oxygen species and their role in Alzheimer disease ': Why does the title not reflect the content of the paragraph? There is no mention of the role of ROS in AD. Also in 3.1. and 3.2. the authors report generic information on ROS with weak references to the CEO. No mention of ROS - neurogenesis interplay in AD
  • Section 4: 4. Mitochondrial dysfunction: generic information about Mitochondria, irrelevant to the topic in question, is provided. Paragraphs 4 and 4.1 : summarize and assemble the 2 paragraphs. In full section 4: No mention of mitochondria - neurogenesis interplay in AD.
  • Figures: The figures must be indicated in the text, not placed at the end of the manuscript.

Figure 3. this figure is showing pathways of neurotoxicity in Alzheimer's disease '. No description of pathways of neurotoxicity in Alzheimer's disease is made in the text. Figure should be deleted or described in the text.

Therefore, I would restructure the entire studio:

  1. AD characteristics: 1. Presentation of the 2 toxic proteins, i.e. Beta-amyloid and Tau; 2. Oxidative stress and mitochondrial dysfunction in AD
  2. Neurogenesis in a healthy and AD context
  3. AD alters the neurogenic process
  4. How Ox stress and mito dysfunction participate in the neurogenic process (if there is any information about it!!! Information that, at the moment, is missing in the current version of the Review)

  • Minor points: English certainly needs to be revised

Author Response

We would like to thank the reviewer for the thoughtful comments and helpful suggestions.

Reviewer #5: The Study by Essa et al 'Neurogenesis, Mitochondrial Dysfunction and Oxidative Stress in Alzheimer Disease' should have neurogenesis in AD as a pivotal topic. 

Major concerns:

Q1.   The authors write in the Abstract: "In the review, we will discuss the recent approaches to neurogenesis in Alzheimer disease and update the development of therapeutic methods.", but honestly, I do not find this sentence to match the content of the manuscript.

Response: We appreciate the reviewer’s concern. We added the development of therapeutic methods as mentioned above.  

Q2. This study intends to treat the relationship between AD and neurogenesis and how this neurodegenerative disease alters the neurogenic process, the title 'Neurogenesis, Mitochondrial Dysfunction and Oxidative Stress in Alzheimer Disease' is rather equivocal. So, it needs to be changed.

Response: We agree with the reviewer. Accordingly, we changed our manuscript’s title as “Implication of Adult Hippocampal Neurogenesis in Alzheimer’s Disease and Potential Therapeutic Approaches”.

Q3.    A meticulous and massive revision work is necessary and urgent. Before detailing changes to be made, I want to point out to the authors that many, many data reported lack bibliographic references. Absolutely to be included. E.g. in #paragraph 4.1 there is a lot of information reported between refs 57 and 58 that are orphans, without a paternity, i.e. without bibliographical references!!! But it is just an example!

Response: We appreciate the reviewer’s comment and we have updated our references in our updated manuscript.

Q4.   The design of the study is confusing. First, I ask the authors: What is the topic they intend to address? According to what is reported in the Abstract, it seems that the spotlight is on neurogenesis in AD. Can the authors confirm this? If Yes, the title must be revised !!! After reading the Title “Neurogenesis, Mitochondrial Dysfunction and Oxidative Stress in Alzheimer Disease” I thought that these 3 different topics were covered in the Review. The Introduction reads 'Here, we briefly go over the literature currently available about the relationship between AD and neurogenesis with an emphasis on the relationship between mitochondrial dysfunction and the effects of oxidative stress on neurogenesis.' But there is no trace of what was planned in the introduction in Sections 4 and 5.

Response: We appreciate the reviewer’s concern. The topic we want to review is neurogenesis in Alzheimer disease and the most impactful factors affecting it along with the most recent therapeutic approaches towards neurogenesis in AD. In agreement with the reviewer’s comment, the introduction has been updated, the sections order has been updated in the revised manuscript, figures has been updated and the title has been changed.

Q5.     In paragraph 2.1, the authors first mention tau protein. They write: 'studies have already shown evidence of Musashi1 interaction with tau', but they do not introduce the tau protein to the reader, who does not necessarily know AD. The same applies to the accumulation of Ab. The authors write '… highlighted the accumulation of extracellular Aβ as the largest likely contributor to reduced neurogenesis in AD.', but again they do not consider that a reader interested in neurogenesis, but not Alzheimer's, may not know the Abeta's role in AD. Therefore, with the risk of being repetitive, it is necessary to 'present' the disease to the 'ignorant' reader!!! I too, when I write Reviews on AD, I ask myself the problem of giving redundant information, but I realize that, even if brief, certain information on the characteristics of AD must be provided, if anything in such a way as to guide it towards the topic of interest.

Response: in our review we reviewed the Amyloid beta pathway and how it is formed, and we explained the amyloidogenic and the non-amyloidogenic pathway. However, we lacked the description of Tau pathology and the neurofibrillay tangles formation, in our revised manuscript we added this, “Alzheimer disease pathogenesis is defined by two major mechanisms, aggregation of intracellular tau proteins [1] and accumulation of misfolded Amyloid beta proteins. The gene responsible for encoding amyloid precursor protein is located on chromosome 21, and once produced the amyloid precursor protein undergoes cleavage by two distinct pathways, the amyloidogenic pathway where it gets cleaved by b-secretase and yields a long soluble extracellular fragment (sAPPβ) and a membrane bound c-terminal peptide termed (β-CTF or C99) containing Aβ which is released after subsequent cleavage by γ-secretase as well as APP intracellular domain (AICD). Alternatively, for the non-amyloidogenic pathway, initial cleavage by α-secretase yields a slightly shorter soluble extracellular fragment termed (sAPPα) and a C-terminal membrane bound peptide termed (α-CTF or C83) which after subsequent cleavage by γ-secretase yields an extracellular small peptide called p3 of unknown function plus AICD [2]. AD is also characterized by Tau protein (τ) accumulation and neurofibrillary tangles (NFTs) formation [3]. Tau is a microtubule associated protein that, when hyperphosphorylated in Alzheimer disease it forms paired protein filaments in the cell cytoplasm termed neurofibrillarly tangles [4].”

Q6.     Also, in paragraph 2.2 'Is it Aβ accumulation or APP expression affecting neurogenesis?', authors mention cystatin C. Again: the authors assume that cystatin C is known to the reader as the concept of oxidative stress may be known since you study it on textbooks! If we go into the experimental detail, it is necessary to accompany the reader - whatever he/she is - to understand the study.

Response: We agree with the reviewer. Accordingly, we added this to describe Cystatin C to the reader, “Cystatin C (CysC) is a protease inhibitor originally identified in cerebrospinal fluid [5]. In AD, it has been found that CysC co-localize with Aβ [6]. In multiple studies it has been demonstrated that co-localization of Cystatin C with Aβ in vessel walls lead to significantly more hemorrhagic events [7]

Q7.     I would certainly deepen paragraph 2.3 'Neurogenesis as a therapeutic target in Alzheimer's disease'. The steps of the research in this sense and if this type of approach has passed from the laboratory to the clinic it would be very important to investigate.

Response: Thank you for the comment and we have enhanced the therapeutic approaches section in our revised manuscript. We supplemented the section with this part to better review the approaches towards enhancing neurogenesis in AD and other neurodegenerative disease, “Another approach to upregulate neurogenesis was to decrease neuroinflammation, which in turn increased neurogenesis. This was achieved using Minocycline, a known broad-spectrum antibiotic and anti-inflammatory drug [8]. They described how Minocycline was able to decrease neuroinflammation in the hippocampus by deactivating microglial activation which is known to play a role in neurodegenerative diseases like Alzheimer disease [9]. Similarly, Chakrabarti et al have described the use of retinoid acid to enhance neurogenesis in AD patients [10]. Retinoid acid is an active natural and synthetic derivative of vitamin A. As described by Ding et al, it also attenuates microglia activation in APP/PS1 mice. As well as it decreased Aβ plaque accumulation and it significantly recovered deficits of learning and memory [11]. Exercise has also been described to increase cell proliferation and promote neurogenesis in adult mouse dentate gyrus [12]. In another follow up study, they described that exercise not only promote neurogenesis but also enhanced spatial learning and long term potentiation of the dentate gyrus [13]. Several literatures have also mentioned that exercise exerts its effect on neurogenesis with brain derived neurotrophic factor (BDNF) [14] which plays a key role in regulation of neurogenesis whether it is proliferation, differentiation, maturation or plasticity [15]. The look up for therapeutic intervention to enhance neurogenesis and alleviate the effect of neuroinflammation on neurogenesis is a still ongoing challenge.”

Q8.     As regards Section 3. '3. Reactive oxygen species and their role in Alzheimer disease ': Why does the title not reflect the content of the paragraph? There is no mention of the role of ROS in AD. Also, in 3.1. and 3.2. the authors report generic information on ROS with weak references to the CEO. No mention of ROS - neurogenesis interplay in AD

Response: We agree with the reviewer. Accordingly, we have combined the reactive oxygen species section to the mitochondrial dysfunction section as advised by the reviewer.

Q9.     Section 4: 4. Mitochondrial dysfunction: generic information about Mitochondria, irrelevant to the topic in question, is provided. Paragraphs 4 and 4.1: summarize and assemble the 2 paragraphs. In full section 4: No mention of mitochondria - neurogenesis interplay in AD.

Response: We agree with the reviewer. Accordingly, we have changed the order of the sections in our revised manuscript and most importantly we tried to give bullet points about mitochondria and cut down on the generic information. As well as we have added this part to the revised manuscript to make a better connection between mitochondrial dysfunction and neurogenesis: “Mitochondrial dysfunction impairs neurogenesis not only in Alzheimer disease but in other neurodegenerative diseases [16]. As it shows, Khacko et al have described that mitochondrial dysfunction results in the deficit of neurogenesis in both contexts of brain embryonic development. In another study where they targeted the basic-helix-loop-helix transcription factor (bHLH) neurod1, which has been proven to be crucial for newborn granule cells to successfully mature and survive in the healthy adult brain [17]. Richtein et al have described that overexpression of neurod1 not only increase neurogenesis compared to control mice, but also significantly increase mitochondrial mass [18], which further proves the fact that mitochondrial rescue can add to increase in neurogenesis and that mitochondria is definitely a possible target for rescuing neurogenesis.” 

Q10.     Figures: The figures must be indicated in the text, not placed at the end of the manuscript.

Response: the figures have been moved and indicated in the text as advised.

Q11.     Figure 3. this figure is showing pathways of neurotoxicity in Alzheimer's disease '. No description of pathways of neurotoxicity in Alzheimer's disease is made in the text. Figure should be deleted or described in the text.

Response: The description of the pathways has been added to Figure 3 legend in the revised manuscript.

Q12.     Therefore, I would restructure the entire studio:

AD characteristics: 1. Presentation of the 2 toxic proteins, i.e. Beta-amyloid and Tau; 2. Oxidative stress and mitochondrial dysfunction in AD

Neurogenesis in a healthy and AD context

AD alters the neurogenic process

How Ox stress and mito dysfunction participate in the neurogenic process (if there is any information about it!!! Information that, now, is missing in the current version of the Review)

Response: We agree with the reviewer’s point. Therefore, the structure is switched in our revised manuscript to offer better scientific flow to the reader as the following;

  1. presentation of the 2 toxic proteins Aβ and tau in Alzheimer disease.
  2. Oxidative stress and mitochondrial dysfunction in AD
  3. Introduction to neurogenesis and neuroegenesis in a healthy context
  4. How AD alters neurogenesis including how Oxidative stress and mitochondrial dysfunctions plays a role in neurogenesis in AD.
  5. Therapeutic approaches to upregulate the neuroegenesis process.

Minor concern: English certainly needs to be revised.

Response: We agree with the reviewer. English and sentence structures is revised in our updated manuscript. Including, decreasing the complexity of some sentences to better deliver information to the reader as well as grammatical revision by several native English speakers as we acknowleged in the manuscript.

References in the reply:

  1. Fu, H.; Rodriguez, G.A.; Herman, M.; Emrani, S.; Nahmani, E.; Barrett, G.; Figueroa, H.Y.; Goldberg, E.; Hussaini, S.A.; Duff, K.E. Tau Pathology Induces Excitatory Neuron Loss, Grid Cell Dysfunction, and Spatial Memory Deficits Reminiscent of Early Alzheimer's Disease. Neuron 2017, 93, 533-541 e535, doi:10.1016/j.neuron.2016.12.023.
  2. Bayer, T.A.; Wirths, O.; Majtenyi, K.; Hartmann, T.; Multhaup, G.; Beyreuther, K.; Czech, C. Key factors in Alzheimer's disease: beta-amyloid precursor protein processing, metabolism and intraneuronal transport. Brain Pathol 2001, 11, 1-11, doi:10.1111/j.1750-3639.2001.tb00376.x.
  3. Ittner, L.M.; Ke, Y.D.; Delerue, F.; Bi, M.; Gladbach, A.; van Eersel, J.; Wolfing, H.; Chieng, B.C.; Christie, M.J.; Napier, I.A.; et al. Dendritic function of tau mediates amyloid-beta toxicity in Alzheimer's disease mouse models. Cell 2010, 142, 387-397, doi:10.1016/j.cell.2010.06.036.
  4. Tiwari, S.; Atluri, V.; Kaushik, A.; Yndart, A.; Nair, M. Alzheimer's disease: pathogenesis, diagnostics, and therapeutics. Int J Nanomedicine 2019, 14, 5541-5554, doi:10.2147/IJN.S200490.
  5. Mathews, P.M.; Levy, E. Cystatin C in aging and in Alzheimer's disease. Ageing Res Rev 2016, 32, 38-50, doi:10.1016/j.arr.2016.06.003.
  6. Levy, E.; Sastre, M.; Kumar, A.; Gallo, G.; Piccardo, P.; Ghetti, B.; Tagliavini, F. Codeposition of cystatin C with amyloid-beta protein in the brain of Alzheimer disease patients. J Neuropathol Exp Neurol 2001, 60, 94-104, doi:10.1093/jnen/60.1.94.
  7. Itoh, Y.; Yamada, M.; Hayakawa, M.; Otomo, E.; Miyatake, T. Cerebral amyloid angiopathy: a significant cause of cerebellar as well as lobar cerebral hemorrhage in the elderly. J Neurol Sci 1993, 116, 135-141, doi:10.1016/0022-510x(93)90317-r.
  8. Wadhwa, M.; Prabhakar, A.; Ray, K.; Roy, K.; Kumari, P.; Jha, P.K.; Kishore, K.; Kumar, S.; Panjwani, U. Inhibiting the microglia activation improves the spatial memory and adult neurogenesis in rat hippocampus during 48 h of sleep deprivation. J Neuroinflammation 2017, 14, 222, doi:10.1186/s12974-017-0998-z.
  9. Lue, L.F.; Walker, D.G.; Rogers, J. Modeling microglial activation in Alzheimer's disease with human postmortem microglial cultures. Neurobiol Aging 2001, 22, 945-956, doi:10.1016/s0197-4580(01)00311-6.
  10. Chakrabarti, M.; McDonald, A.J.; Will Reed, J.; Moss, M.A.; Das, B.C.; Ray, S.K. Molecular Signaling Mechanisms of Natural and Synthetic Retinoids for Inhibition of Pathogenesis in Alzheimer's Disease. Journal of Alzheimer's disease : JAD 2016, 50, 335-352, doi:10.3233/JAD-150450.
  11. Ding, Y.; Qiao, A.; Wang, Z.; Goodwin, J.S.; Lee, E.S.; Block, M.L.; Allsbrook, M.; McDonald, M.P.; Fan, G.H. Retinoic acid attenuates beta-amyloid deposition and rescues memory deficits in an Alzheimer's disease transgenic mouse model. J Neurosci 2008, 28, 11622-11634, doi:10.1523/JNEUROSCI.3153-08.2008.
  12. van Praag, H.; Kempermann, G.; Gage, F.H. Running increases cell proliferation and neurogenesis in the adult mouse dentate gyrus. Nat Neurosci 1999, 2, 266-270, doi:10.1038/6368.
  13. van Praag, H.; Christie, B.R.; Sejnowski, T.J.; Gage, F.H. Running enhances neurogenesis, learning, and long-term potentiation in mice. Proceedings of the National Academy of Sciences of the United States of America 1999, 96, 13427-13431, doi:10.1073/pnas.96.23.13427.
  14. Liu, P.Z.; Nusslock, R. Exercise-Mediated Neurogenesis in the Hippocampus via BDNF. Front Neurosci 2018, 12, 52, doi:10.3389/fnins.2018.00052.
  15. Bath, K.G.; Akins, M.R.; Lee, F.S. BDNF control of adult SVZ neurogenesis. Dev Psychobiol 2012, 54, 578-589, doi:10.1002/dev.20546.
  16. Khacho, M.; Clark, A.; Svoboda, D.S.; MacLaurin, J.G.; Lagace, D.C.; Park, D.S.; Slack, R.S. Mitochondrial dysfunction underlies cognitive defects as a result of neural stem cell depletion and impaired neurogenesis. Hum Mol Genet 2017, 26, 3327-3341, doi:10.1093/hmg/ddx217.
  17. Gao, Z.; Ure, K.; Ables, J.L.; Lagace, D.C.; Nave, K.A.; Goebbels, S.; Eisch, A.J.; Hsieh, J. Neurod1 is essential for the survival and maturation of adult-born neurons. Nat Neurosci 2009, 12, 1090-1092, doi:10.1038/nn.2385.
  18. Richetin, K.; Moulis, M.; Millet, A.; Arrazola, M.S.; Andraini, T.; Hua, J.; Davezac, N.; Roybon, L.; Belenguer, P.; Miquel, M.C.; et al. Amplifying mitochondrial function rescues adult neurogenesis in a mouse model of Alzheimer's disease. Neurobiol Dis 2017, 102, 113-124, doi:10.1016/j.nbd.2017.03.002.

Round 2

Reviewer 3 Report

In this revision, the authors have addressed my concerns to some extent. However, there are still aspects that needs to be fixed.

In the introduction, the author stated that “we briefly go over the literature currently available about the relationship between AD and neurogenesis with an emphasis on the relationship between mitochondrial dysfunction and the effects of oxidative stress on neurogenesis”.  However, throughout the manuscript, the author used only a few sentences to address the relationship between relationship between mitochondrial dysfunction and the effects of oxidative stress on neurogenesis. The reviewer did not see how the authors emphasize such a relationship.

The author also stated that “We will also review the available literature on targeting oxidative stress as a possible treatment target for AD”.  However, the authors only talk about neurogenesis as therapeutic targets.

Thus, the reviewer requests the authors to revise the manuscript carefully and systemically.

Author Response

We would like to thank the reviewer for the thoughtful comments and helpful suggestions.

Reviewer #3: In this revision, the authors have addressed my concerns to some extent. However, there are still aspects that needs to be fixed.

Minor concerns:

Q1. In the introduction, the author stated that “we briefly go over the literature currently available about the relationship between AD and neurogenesis with an emphasis on the relationship between mitochondrial dysfunction and the effects of oxidative stress on neurogenesis”.  However, throughout the manuscript, the author used only a few sentences to address the relationship between relationship between mitochondrial dysfunction and the effects of oxidative stress on neurogenesis. The reviewer did not see how the authors emphasize such a relationship.

Response: We agree with the reviewer’s comment. Accordingly, we added this part to our revised manuscript; “Consequently, suitable levels of ROS play a role in maintaining appropriate levels of neurogenesis. Mitochondria are implicated as regulators of neurogenesis, generate ROS, and ROS, when over accumulated, lead to mitochondrial dysfunction. Therefore, ROS and oxidative stress may be implicated as regulators of neurogenesis [1]. Studies indicated that reduced cellular redox status tend to favor cell proliferation whereas oxidized environments favor cell differentiation under physiological conditions [2]. Under pathological conditions, however, such as AD, mitochondrial dysfunction may tip the balance as ROS accumulate leading to oxidative stress and further impairments in mitochondrial function [3]. The most common contributor to oxidative stress is superoxide anion (O2-) [4] which is neutralized by super oxide dismutase (SOD) [5]. In a study investigating neurogenesis in mice exposed to radiation, those lacking extracellular SOD exhibited reduced numbers of BrdU+/NeuN+ cells in the subgranular zone of the hippocampus in knockout mice compared to wildtype mice [6]. In fact, regardless of the subcellular localization, SOD deficiency leads to reductions in neurogenesis in the SGZ of the hippocampus. Interestingly, however, in response to radiation, SOD deficient mice were able to maintain comparable levels of neurogenesis. [7].”

Q2. The author also stated that “We will also review the available literature on targeting oxidative stress as a possible treatment target for AD”.  However, the authors only talk about neurogenesis as therapeutic targets.

Response: Thank you for your comment. Agreeing with your statement, we added this to our revised manuscript; “Since the relationship between excessive oxidative stress and inhibited neurogenesis has been established, ROS is a new emerging therapeutics target in AD.  Massieu et al, reported that the neuronal damage induced by Aβ was prevented in rats treated with Vitamin E [8]. Vitamin E exhibits antioxidant effects [9]. In another study, the higher dietary intake of Vitamin E was associated with lower risk of AD [10]. Vitamin C which has antioxidant effects [11] was also shown to decrease behavioral abnormalities in AD mouse model [12]. Another possible antioxidant for treatment of AD is curcumin, a curry spice, which exhibits antioxidant effects [13]. Curcumin exerts its effects by decreasing the low-density lipoprotein oxidation and the free radicals that eventually lead to neurodegeneration [14]. Using Tg2576 mice model of AD, Curcumin reduced the plaque burden of insoluble and soluble Aβ which further shows its promise in the prevention of AD [15].”

References in the reply:

  1. Rego, A.C.; Oliveira, C.R. Mitochondrial dysfunction and reactive oxygen species in excitotoxicity and apoptosis: implications for the pathogenesis of neurodegenerative diseases. Neurochem Res 2003, 28, 1563-1574, doi:10.1023/a:1025682611389.
  2. Schafer, F.Q.; Buettner, G.R. Redox environment of the cell as viewed through the redox state of the glutathione disulfide/glutathione couple. Free radical biology & medicine 2001, 30, 1191-1212, doi:10.1016/s0891-5849(01)00480-4.
  3. Misrani, A.; Tabassum, S.; Yang, L. Mitochondrial Dysfunction and Oxidative Stress in Alzheimer's Disease. Front Aging Neurosci 2021, 13, 617588, doi:10.3389/fnagi.2021.617588.
  4. Chen, Y.; Azad, M.B.; Gibson, S.B. Superoxide is the major reactive oxygen species regulating autophagy. Cell Death Differ 2009, 16, 1040-1052, doi:10.1038/cdd.2009.49.
  5. Gray, B.; Carmichael, A.J. Kinetics of superoxide scavenging by dismutase enzymes and manganese mimics determined by electron spin resonance. Biochem J 1992, 281 ( Pt 3), 795-802, doi:10.1042/bj2810795.
  6. Rola, R.; Zou, Y.; Huang, T.T.; Fishman, K.; Baure, J.; Rosi, S.; Milliken, H.; Limoli, C.L.; Fike, J.R. Lack of extracellular superoxide dismutase (EC-SOD) in the microenvironment impacts radiation-induced changes in neurogenesis. Free radical biology & medicine 2007, 42, 1133-1145; discussion 1131-1132, doi:10.1016/j.freeradbiomed.2007.01.020.
  7. Huang, T.T.; Zou, Y.; Corniola, R. Oxidative stress and adult neurogenesis--effects of radiation and superoxide dismutase deficiency. Semin Cell Dev Biol 2012, 23, 738-744, doi:10.1016/j.semcdb.2012.04.003.
  8. Montiel, T.; Quiroz-Baez, R.; Massieu, L.; Arias, C. Role of oxidative stress on beta-amyloid neurotoxicity elicited during impairment of energy metabolism in the hippocampus: protection by antioxidants. Exp Neurol 2006, 200, 496-508, doi:10.1016/j.expneurol.2006.02.126.
  9. Traber, M.G.; Atkinson, J. Vitamin E, antioxidant and nothing more. Free radical biology & medicine 2007, 43, 4-15, doi:10.1016/j.freeradbiomed.2007.03.024.
  10. Devore, E.E.; Grodstein, F.; van Rooij, F.J.; Hofman, A.; Stampfer, M.J.; Witteman, J.C.; Breteler, M.M. Dietary antioxidants and long-term risk of dementia. Arch Neurol 2010, 67, 819-825, doi:10.1001/archneurol.2010.144.
  11. Padayatty, S.J.; Katz, A.; Wang, Y.; Eck, P.; Kwon, O.; Lee, J.H.; Chen, S.; Corpe, C.; Dutta, A.; Dutta, S.K.; et al. Vitamin C as an antioxidant: evaluation of its role in disease prevention. J Am Coll Nutr 2003, 22, 18-35, doi:10.1080/07315724.2003.10719272.
  12. Murakami, K.; Murata, N.; Ozawa, Y.; Kinoshita, N.; Irie, K.; Shirasawa, T.; Shimizu, T. Vitamin C restores behavioral deficits and amyloid-beta oligomerization without affecting plaque formation in a mouse model of Alzheimer's disease. Journal of Alzheimer's disease : JAD 2011, 26, 7-18, doi:10.3233/JAD-2011-101971.
  13. Ak, T.; Gulcin, I. Antioxidant and radical scavenging properties of curcumin. Chem Biol Interact 2008, 174, 27-37, doi:10.1016/j.cbi.2008.05.003.
  14. Kim, G.Y.; Kim, K.H.; Lee, S.H.; Yoon, M.S.; Lee, H.J.; Moon, D.O.; Lee, C.M.; Ahn, S.C.; Park, Y.C.; Park, Y.M. Curcumin inhibits immunostimulatory function of dendritic cells: MAPKs and translocation of NF-kappa B as potential targets. J Immunol 2005, 174, 8116-8124, doi:10.4049/jimmunol.174.12.8116.
  15. Lim, G.P.; Chu, T.; Yang, F.; Beech, W.; Frautschy, S.A.; Cole, G.M. The curry spice curcumin reduces oxidative damage and amyloid pathology in an Alzheimer transgenic mouse. J Neurosci 2001, 21, 8370-8377.

Reviewer 5 Report

The revision work done by the authors is satisfying.
I believe the study deserves publication now.

Author Response

.